# A Proteomic Screen to Unravel the Molecular Pathways Associated with Warfarin-Induced or TNAP-Inhibited Arterial Calcification in Rats

**DOI:** 10.3390/ijms24043657

**Published:** 2023-02-11

**Authors:** Britt Opdebeeck, Ellen Neven, Stuart Maudsley, Hanne Leysen, Deborah Walter, Hilde Geryl, Patrick C. D’Haese, Anja Verhulst

**Affiliations:** 1Laboratory of Pathophysiology, Department of Biomedical Sciences, University of Antwerp, 2000 Antwerpen, Belgium; 2Receptor Biology Lab, Department of Biomedical Sciences, University of Antwerp, 2000 Antwerpen, Belgium

**Keywords:** arterial media calcification, tissue non-specific alkaline phosphatase, inflammation, lipid/glucose homeostasis, mitochondrial pathway

## Abstract

Arterial media calcification refers to the pathological deposition of calcium phosphate crystals in the arterial wall. This pathology is a common and life-threatening complication in chronic kidney disease, diabetes and osteoporosis patients. Recently, we reported that the use of a TNAP inhibitor, SBI-425, attenuated arterial media calcification in a warfarin rat model. Employing a high-dimensionality unbiased proteomic approach, we also investigated the molecular signaling events associated with blocking arterial calcification through SBI-425 dosing. The remedial actions of SBI-425 were strongly associated with (i) a significant downregulation of inflammatory (acute phase response signaling) and steroid/glucose nuclear receptor signaling (LXR/RXR signaling) pathways and (ii) an upregulation of mitochondrial metabolic pathways (TCA cycle II and Fatty Acid β-oxidation I). Interestingly, we previously demonstrated that uremic toxin-induced arterial calcification contributes to the activation of the acute phase response signaling pathway. Therefore, both studies suggest a strong link between acute phase response signaling and arterial calcification across different conditions. The identification of therapeutic targets in these molecular signaling pathways may pave the way to novel therapies against the development of arterial media calcification.

## 1. Introduction

The high-dimensionality appreciation of human physiological and pathological mechanisms is continuously improving, leading to an elevation in lifespan throughout the world. Aging-related diseases are now a prominent feature in our society [1,2,3]. Chronic kidney disease (CKD), diabetes mellitus and osteoporosis are three examples of highly prevalent aging-related disorders that demonstrate multiple intertwined gerontological pathomechanisms. Patients afflicted with these conditions unfortunately often die from severe cardiovascular complications, including arterial media calcification, rather than from the primary disease itself [4,5,6]. During arterial media calcification, calcium phosphate crystals (also called hydroxyapatite) precipitate in the medial layer of the arterial vessel wall, thereby promoting arterial stiffness which in turn increases systolic blood pressure and engenders an elevated risk for left ventricular hypertrophy, impaired peripheral blood circulation and myocardial infarction [7]. Arterial media calcification is characterized by (i) a passive precipitation of calcium phosphate mineral and (ii) an active cell-regulated process in which vascular smooth muscle cells (VSMCs) undergo transdifferentiation towards bone-like cells [8]. In addition, the apoptosis of VSMCs is frequently evident during arterial media calcification, leading to the generation of apoptotic bodies which are perfect nidi for calcium phosphate crystal aggregation and growth [9,10]. Furthermore, reductions in calcification inhibitors, with a concomitant upregulation of pro-calcification factors, are often observed in these patients [8]. 

The anti-coagulant warfarin, when given at a high dosage (four–six times the normal therapeutic dosage employed for anti-coagulation) to rats stimulates a gradual development of calcification in the medial layer of the arterial wall over six weeks [11]. Warfarin blocks vitamin K recycling, leading to less vitamin K-dependent activation of the important calcification inhibitor matrix gla protein (MGP) [12]. 

Tissue non-specific alkaline phosphatase (TNAP) represents one of the aforementioned pro-calcification factors as it facilitates the hydrolysis of the calcification inhibitor pyrophosphate into inorganic phosphate [13]. Multiple studies have shown that TNAP overexpression in mice leads to overt arterial calcification [14,15,16]. We recently demonstrated that treatment with a tissue non-specific alkaline phosphatase (TNAP) inhibitor SBI-425 significantly inhibited warfarin-induced arterial media calcification in rats [17]. In the present study, using a high-dimensionality unbiased proteomic approach, we investigated the molecular nature of the signaling events associated with blocking arterial calcification through the TNAP inhibitor SBI-425.

## 2. Results

### 2.1. Proteome Profile of the Aortic Tissue of SBI-425 versus Vehicle-Treated Animals Exposed to Warfarin

In rats, the decrease in aortic calcification by TNAP inhibition resulted in the differential up- and downregulation of 155 and 159 proteins versus warfarin induced aortic calcification, respectively (see Appendix A). Table 1 gives the top 10 most differentially up- and down-regulated proteins in the aortic tissue of SBI-425-treated rats versus vehicle.

### 2.2. Relative Quantification of Fat Uncoupling Protein 1 in the Aortic Tissue

Western blot analysis of mitochondrial brown fat uncoupling protein 1 (UCP1), the third most up-regulated protein in Table 1, confirmed an increased expression of this protein, although not significant, in the aortic tissue of animals treated with SBI-425 versus vehicle ones. Figure 1 shows the relative quantification of a UCP1 western blot based on total protein normalization.

### 2.3. Decrease in Aortic Calcification by TNAP Inhibition Associates with Lipid/Glucose Homeostasis and Inflammation Pathways

An IPA-based canonical signaling pathway analysis was performed to link and coherently cluster the differentially expressed proteins to mechanistic signaling pathways. Both the inflammation pathway ‘Acute phase response signaling’ and the lipid/glucose homeostasis regulation pathway ‘LXR/RXR signaling’ were significantly populated, predominantly by downregulated proteins (Figure 2). 

Furthermore, the predicted pathway z-score for the acute phase response signaling and LXR/RXR signaling were −0.7 and −2.67, respectively (Appendix A). An opposite effect is observed with regard to metabolic pathway regulation through TNAP inhibition (Figure 3).

In the aortic tissue of SBI-425-treated rats, the metabolic pathways ‘TCA cycle II’ and ‘Fatty Acid β-oxidation I’ were populated predominantly by upregulated proteins and in both cases showed positive predicted metabolic pathway z-scores of 3.162 (Appendix A). In addition, Figure 4 compares the relative expression profiles of the proteins in our proteome dataset to their effect on the respective pathway. This was carried out for ‘Acute phase response signaling’, ‘LXR/RXR signaling’, ‘TCA cycle II’ and ‘Fatty Acid β-oxidation I’.

### 2.4. Decrease in Aortic Calcification by TNAP Inhibition Associates with Anti-Hyperglycemic Events

Using the IPA-based Upstream analysis suite, the identification of upstream chemical and protein factors that are responsible for the up- or downregulation of 314 proteins in the SBI-425 proteome was performed. This analysis revealed a strong link between noncalcified SBI-425-treated vessels and (i) the upregulation of insulin-sensitizing proteins (i.e., PPAR signaling, insulin receptor) and (ii) chemical insulin sensitizers (i.e., rosiglitazone and troglitazone) (Figure 5).

## 3. Discussion

There is a pressing need for high-dimensionality analytical studies investigating novel therapeutic targets to treat arterial media calcification as this pathology significantly contributes to severe cardiovascular complications and increased mortality risk in the elderly and patients with CKD, diabetes and osteoporosis [4,5,6]. For this reason, a quantitative proteomic discovery/screening approach would be helpful to unravel important pathways that are involved in the pathogenic mechanism underlying the disease. In this study, arterial media calcification was induced in rats by exposing them to high warfarin doses, such that the calcification inhibitor matrix gla protein was inactivated [12]. As described in a previous publication, TNAP inhibitor SBI-425 efficiently inhibits the presence of arterial media calcification in these warfarin exposed rats [17]. However, here, we linked mechanistic signaling pathways to 314 differentially expressed proteins in the noncalcified aortic tissue of SBI-425-treated rats exposed to warfarin. A decrease in the inflammation pathway ‘Acute phase response signaling’ and the lipid/glucose homeostasis pathway ‘LXR/RXR signaling’ and an increase in the metabolic mitochondrial pathways ‘TCA cycle II’ and ‘Fatty Acid β-oxidation I’ was found. Interestingly, in line with these results, we reported that protein-bound uremic toxins indoxyl sulfate and p-cresyl sulfate induce arterial media calcification in CKD rats through the induction of the ‘Acute phase response signaling’ pathway [18]. Additionally, consistent with our results, Takehito Okui et al., using tandem mass tagging (TMT) proteomics, identified a role for fatty acid oxidation and mitochondrial TCA cycle pathways in calcified primary human coronary artery smooth muscle cells [19]. Vascular calcification in mice with the premature aging disorder (Hutchinson–Gilford progeria syndrome) is linked to VSMC mitochondrial dysfunction as a result of TNAP overexpression [20]. Moreover, Hsu J.J. et al. showed that LXR beta agonist-mediated VSMC calcification was attenuated by levamisol, a TNAP inhibitor [21], which is in agreement with our proteomic screen. Therefore, TNAP inhibitor SBI-425 treatment might inhibit aortic calcification in our rats by enhancing mitochondrial function and blocking the lipid homeostasis regulator LXR beta. 

Furthermore, the thermogenesis-regulating protein UCP1 was the thirdly most upregulated protein in the aortic proteome of SBI-425-treated rats. A similar trend, although not significant, was found through selective UCP1 western blotting. The research group of Spiegelman et al. showed that bona fide VSMCs can convert into UCP1/PPARG-positive adipocyte-like cells via the overexpression of PRDM16 protein (coregulatory protein controlling brown adipocyte differentiation) and exposure to an adipogenic cocktail (insulin, rosiglitazone and triiodothyronine) [22]. In addition, the proteins UCP1, FABP4, PLIN4, SLC27A1, SIRT2 and COX7A2 were significantly upregulated in non-calcified/SBI-425-treated aortic tissues (see (differential expressed proteins) DEP list in Appendix A), and all of these are markers of adipose tissue [23]. Moreover, physical exercise is linked to regulating thermogenesis in brown adipocyte tissue [24]. Interestingly, several studies have shown a correlation between high-intensity/-volume exercise and arterial calcification [25]. Additionally, (i) marathon runners experience NLRP3 inflammasome-related vascular inflammation [26], in which UCP1 expression may play a role [27], and (ii) a strong correlation between the upregulation of acute phase proteins and performing world-class/Olympic sports has been found [28]. These findings may contribute to a potential explanation for a higher prevalence of arterial calcification in elite athletes. This is in line with the present study, which links arterial media calcification to an up- and downregulation of the acute phase response pathway and thermogenesis, respectively.

Furthermore, curated chemical and genomic/proteomic perturbagen matching analysis—predicting upstream regulators that could be responsible for the observed changes in the arterial proteins linked to SBI-425-induced signaling pathways—provided strong evidence for insulin-sensitizing protein regulators (PPAR signaling, insulin receptor signaling). Interestingly, we previously reported that the induction of CKD related arterial media calcification by protein-bound uremic toxins is correlated with increased glucose levels and insulin resistance [18]. Taken together, these results suggest that the thermogenesis regulator PPAR gamma and insulin receptor signaling activate the mitochondrial pathway ‘TCA cycle II’ and the metabolic pathway ‘Fatty Acid β-oxidation I’ in the noncalcified aortic tissue of SBI-425-treated rats. 

While demonstrating nuanced and relevant details concerning the therapeutic effects of SBI-425, our study of course possesses certain limitations: (i) we cannot exclude whether the up/downregulation of the different pathways in the aortic tissue is directly attributed to SBI-425 treatment or rather due to the elimination of calcification, and (ii) this proteomic screening approach must be seen as a foundation for finding novel mechanisms (e.g., thermogenesis) in the arterial calcification process, bearing in mind that the further validation of potential agonists/antagonists of these pathways to provoke/inhibit arterial media calcification is necessary. Next to this, given the characteristics of the warfarin rat model, which only develops media calcifications, we only focused on this type of arterial calcification. It is well-known, however, that atherosclerotic plaque (intima) calcification is also often present in ageing diseases (i.e., CKD and diabetes). Since studies have reported an association between calcified plaques and an improved event-free survival due to plaque stabilization [29], it would be of value to validate the agonists/antagonists of the above-mentioned pathways in animal models developing either only intima or both intima and media arterial calcification. 

In conclusion, the inhibition of arterial media calcification by TNAP inhibitor SBI-425 resulted into a downregulation of inflammation and lipid/glucose homeostasis pathways while favoring mitochondrial pathways, which could be linked to promoting thermogenesis.

## 4. Materials and Methods

### 4.1. Animal Experimentation 

All animal experiments were performed in accordance with the National Institute of Health’s Guide for the Care and Use of Laboratory Animals 85-23 (1996) and approved by the University of Antwerp Ethics Committee (Permit number: 2017-05). Animals were housed two per cage, exposed to 12-h light/dark cycles and had free access to food and water. To induce vascular calcification, 20 male Wistar rats (200-250g, Iffa Credo, Belgium) were administered a warfarin-containing diet (0.30% warfarin and 0.15% vitamin K1 to prevent lethal bleeding) (synthetic diet from SSNIFF Spezialdiäten, Soest, Germany) for the entire study period and were subjected to the following treatments: (i) vehicle (n = 10) or (ii) 10 mg/kg/day SBI-425 (TNAP inhibitor) (n = 10). SBI-425 was provided by J.L. Millán and synthesized at the Prebys Center for Drug Discovery, Sanford Burnham Prebys Medical Discovery Institute (La Jolla, CA, USA). Vehicle and SBI-425 were administered daily via an i.p. catheter (rounded polyurethane catheter ROPAC-3.5PR, Access technologies, Skokie, IL, USA) from the start of warfarin exposure until the end of the study at week 7, as described in [17]. SBI-425 was dissolved in 1% ethanol, 0.3% sodium hydroxide and 98.97% dextrose (5% dissolved in PBS). Vehicle treatment consisted of the solvent without the addition of SBI-425. To collect aortic tissue for proteomic evaluation, rats were sacrificed by exsanguination via the retro-orbital plexus after anesthesia with 80 mg/kg ketamine (Pfizer, Puurs, Belgium) and 10 mg/kg xylazine (Bayer Animal Health, Monheim, Germany) via intraperitoneal injection.

### 4.2. Mass Spectrometry and Quantitative Proteomics of Aortic Samples 

An unbiased quantitative proteomic approach using iTRAQ (isobaric mass-tag labeling for relative and absolute quantitation) labeling and mass spectrometry (MS) was applied to simultaneously identify and quantify the proteins that are differentially up- or downregulated in aortic samples of SBI-425 versus vehicle -exposed CKD animals. Hereto, the distal part of the abdominal aorta (from gonadal arteries to iliac split) was used. For each group, a selection of four distal abdominal aortas was performed based on the calcium scores found in the proximal part of the same abdominal aorta (from diaphragm to gonadal arteries). Samples with the highest calcium score were chosen from the vehicle group, while those with the lowest calcium score were chosen from the SBI-425 group. The aorta samples were ground completely in the protein extraction buffer (8 M urea, 2 M thiourea, 0.1% SDS in 50 mM triethylammonium bicarbonate solution). The concentrations of the proteins extracted from the aorta were quantified using the RCDC kit (Bio-Rad, Hercules, CA, USA). Equal amounts of proteins from each sample were reduced and alkylated by tris-2-carboxyethyl phosphine and 5-methyl-methanoethiosulphate before trypsin digestion, respectively. The resulting peptides from each sample were labelled using iTRAQ reagents (Sciex, Framingham, MA, USA) following the manufacturer’s instructions. To improve LC-MS/MS proteome coverage, samples were subjected to a 2D-LC fractionation system (Dionex ULTIMATE 3000, ThermoScientific, Waltham, MA, USA). The mixed peptides were first fractionated on a strong cationic exchange chromatography polysulfoethyl aspartamide column (1 mm × 150 mm, (Dionex)) and secondly separated on a nano-LC C18 column (200 Å, 2 μm, 75 μm × 25 cm (Dionex)). The nano-LC was coupled online to a QExactive™-Plus Orbitrap (ThermoScientific) mass spectrometer. The nano-LC eluents were infused to the Orbitrap mass-spectrometer with a capillary at 1.7 KV on a nano-ESI source at a flow rate of 300 nl/min. Data-dependent acquisition in positive ion mode was performed for a selected mass range of 350–1800 m/z at MS1 level (140,000 resolution) and MS2 level (17,500 resolution). The raw data were analyzed by Proteome Discoverer 2.0 Software (ThermoScientific) using Sequest HT as a search engine against the human UniProt/SwissProt database with a threshold of confidence above 99% (false discovery rate less than 1%). The list of identified proteins, containing iTRAQ ratios of expression levels over control samples, was generated.

### 4.3. Bioinformatic Analyses

Proteins identified according to the statistical MS cut-offs described previously were then subsequently used for further bioinformatics analyses [11,18]. To identify the significantly altered proteins, i.e., proteins differentially expressed due to SBI-425 treatment compared to vehicle exposure, raw iTRAQ ratios (SBI-425:vehicle) were first log2 transformed. Following log2 ratio transformation, differentially expressed protein (DEP) lists were created by identifying only proteins that possessed log2-transformed iTRAQ ratios two standard deviations (*p* < 0.05) from the calculated mean background expression variation level. Significant DEP lists (comprising proteins elevated or reduced in their expression in response to SBI-425) were then employed for further bioinformatics deconvolution using diverse informatics platforms including Ingenuity Pathway Analysis (Canonical Signaling Pathway and Upstream Regulator applications; https://www.qiagenbioinformatics.com/products/ingenuity-pathway-analysis/ (accessed on 15 September 2022)), GeneIndexer ([30,31]; https://geneindexer.com/ (accessed on 15 September 2022)), WebGestalt [32] and VennPlex ([33]; https://www.irp.nia.nih.gov/bioinformatics/vennplex.html (accessed on 15 September 2022)).

### 4.4. Western Blot

Protein was isolated from equivalent aorta samples used for mass spectrometry and quantitative proteomics. The aorta samples were ground and incubated on ice in RIPA buffer complemented with phospho- and protease inhibitors for one hour. The concentrations of proteins extracted from the aorta were quantified using the Pierce BCA protein assay (Thermofisher Scientific, Horsham, UK). Equal amounts of proteins (12 µg) from each sample were incubated for 5 min at 95 °C with sample buffer including Laemmli buffer (Bio-Rad Laboratories, Temse, Belgium) and 2.5% β-mercaptoethanol for the denaturation of proteins. Subsequently, proteins were separated by gel-electrophoresis on a 12% Mini-PROTEAN TGC Stain-Free Precast Gel (Bio-Rad Laboratories) at 200 V. Proteins were transferred to a PVDF membrane at 100V for one hour. The membrane was blocked with 5% bovine serum albumin (BSA) in Tris-buffered saline with 0.1% Tween 20 (TSBT) buffer for 1 h. To study the aortic protein expression of uncoupling protein 1 (UCP1), PVDF membrane was incubated overnight at 4 °C with UCP1 primary antibody (dilution 1:1000, Thermofisher Scientific, #PA1-24894, Horsham, UK). Furthermore, the membrane was washed multiple times with TBST and incubated for 1 h at room temperature with horseradish peroxidase (HRP)-conjugated secondary swine anti-rabbit (dilution 1:4000, Dako, P0399, Glostrup, Denmark). StrepTactin-HRP conjugate (dilution 1:10 000) was also added to visualize the unstained protein ladder (Precision Plus Protein unstained standards, Bio-Rad Laboratories, Temse, Belgium). A 5% BSA-TBST solution was used to dilute all antibodies. The membrane was washed multiple times with TBST, and proteins were detected via a Clarity Western enhanced chemiluminescent (ECL) substrate kit (Bio-Rad Laboratories). Relative protein quantification was performed through total lane protein normalization in Image Lab (Bio-rad, https://www.bio-rad.com/en-be/applications-technologies/total-protein-normalization?ID=PODYJQRT8IG9 (accessed on 15 September 2022)).

### 4.5. Statistical Analysis

Statistical comparisons were made by non-parametric testing (GraphPad Prism 9, Graphpad Software Inc., San Diego, CA, USA). To investigate the statistical difference between both groups, a Mann–Whitney U test was applied. A *p*-value < 0.05 was considered significant.

## Figures and Tables

**Figure 1 ijms-24-03657-f001:**
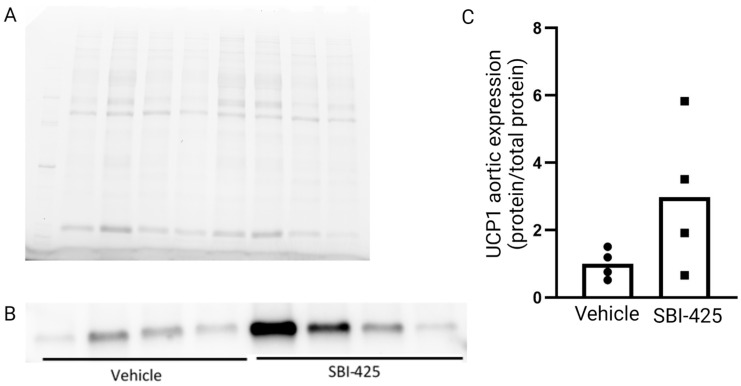
Relative quantification of UCP1 protein expression in the aortic tissue by western blotting. (**A**) Total lane protein quantification, (**B**) expression of UCP1 by western blotting and (**C**) relative quantification of UCP1, based on total lane protein normalization, in vehicle- and SBI-425-treated aortic tissues. Data presented as individual values (dots) and mean (histograms).

**Figure 2 ijms-24-03657-f002:**
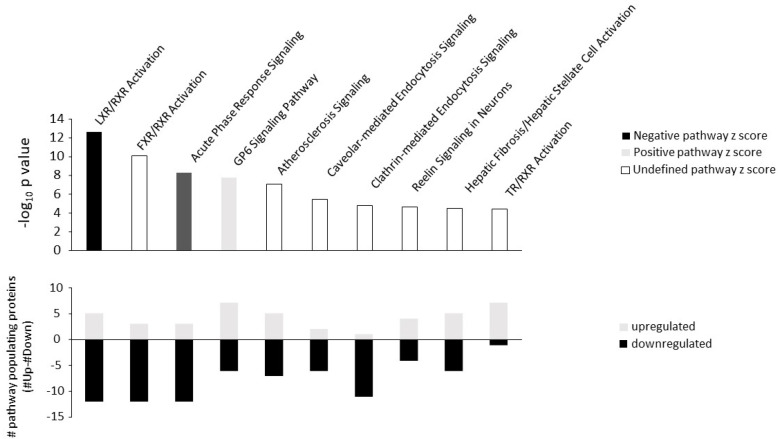
Signaling pathway analysis of the aortic tissue of SBI-425-treated rats versus vehicle. Ingenuity pathway analysis shows the amount of upregulated (grey bars) or downregulated (black bars) proteins in each signaling pathway.

**Figure 3 ijms-24-03657-f003:**
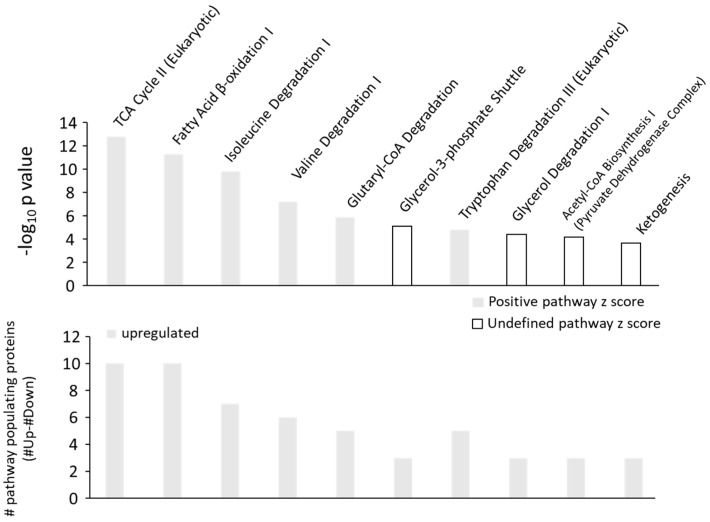
Metabolic pathway analysis of the aortic tissue of SBI-425-treated rats versus vehicle. Ingenuity pathway analysis shows the amount of upregulated (grey bars) proteins in each metabolic pathway.

**Figure 4 ijms-24-03657-f004:**
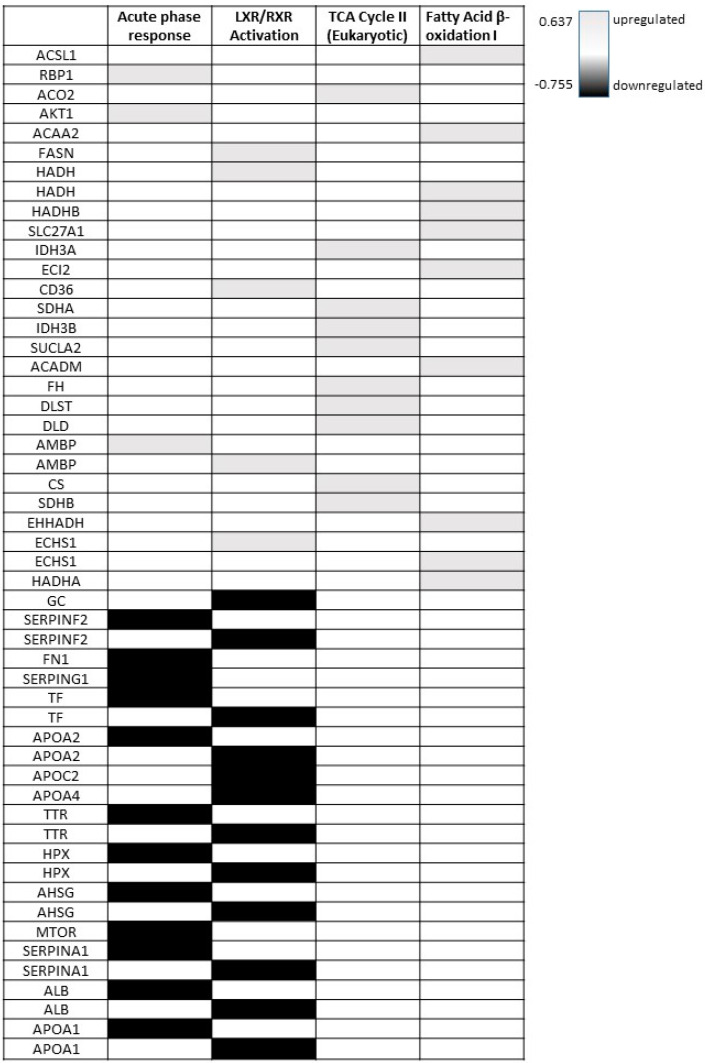
Association of protein responses to inflammatory, lipid/glucose homeostasis and mitochondrial signaling pathways in the aortic tissue of SBI-425-treated rats versus vehicle. A heatmap shows the response (grey = upregulation, black = downregulation) of (differentially expressed proteins) DEPs in the aortic tissue of SBI-425-treated rats to inflammation (acute phase response), lipid/glucose homeostasis regulation (LXR/RXR activation) and mitochondrial (TCA cycle II and Fatty Acid β-oxidation I) pathways.

**Figure 5 ijms-24-03657-f005:**
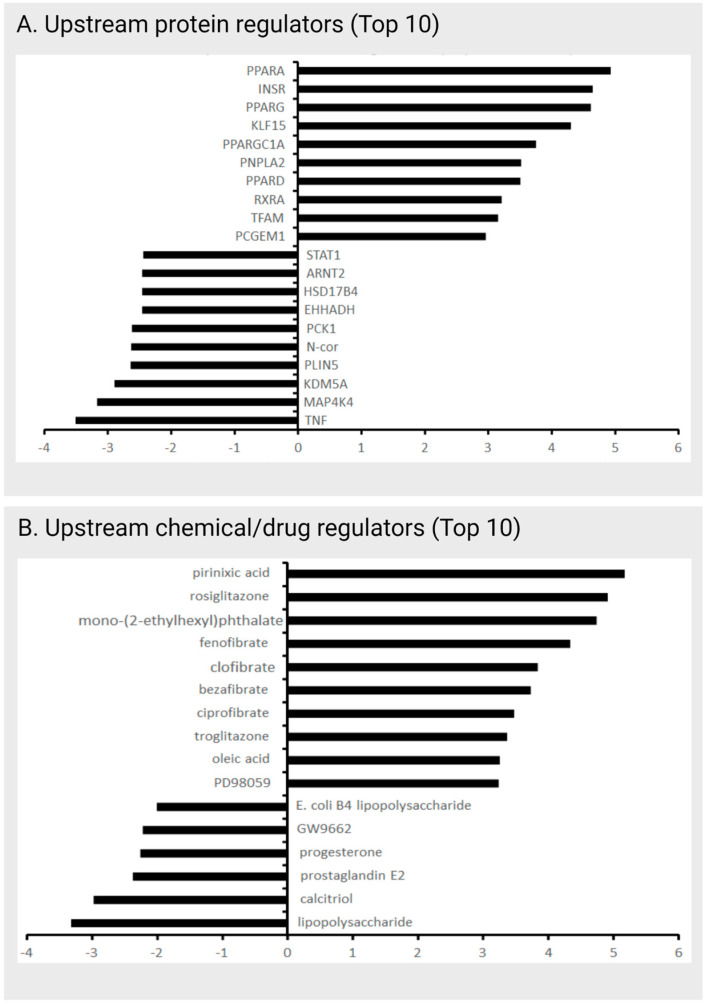
Ingenuity pathway upstream analysis (IPA) of the TNAP inhibitor treatment protein dataset. IPA ascribes a quantified z-score number (ordinate axis) that determines whether an upstream regulator has significantly more “activated” predications (z > 0) than “inhibited” predictions (z < 0). A strong role for altered energy metabolism was observed including upregulation of insulin-sensitizing protein (**A**) (i.e., PPAR signaling, insulin receptor (INSR)) and chemical regulators (**B**) (i.e., rosiglitazone and troglitazone).

**Table 1 ijms-24-03657-t001:** Top 10 differentially up- or downregulated proteins in SBI-425 versus vehicle-treated aortic samples.

Protein Description	Gene Symbol	Log_2_ iTRAQ Ratio
Acyl-CoA desaturase	SCD1	0.718574
Histidine-rich calcium binding protein, isoform CRA_b	HRC	0.710522
Mitochondrial brown fat uncoupling protein 1	UCP1	0.66522
Collagen alpha-1(I) chain	COL1A1	0.663793
CCR4-NOT transcription complex, subunit 1	CNOT1	0.657795
Insulin receptor substrate 1	IRS1	0.655246
Mitochondrial uncoupling protein 3	UCP3	0.654558
Adenylosuccinate synthetase isozyme 1	ADSSL1	0.650851
Fatty acid-binding protein, adipocyte	FABP4	0.648864
Long-chain-fatty-acid-CoA ligase 1	ACSL1	0.637461
Seminal vesicle secretory protein 4	SVS4	−3.16592
Seminal vesicle secretory protein 6	SVS6	−1.85127
Seminal vesicle secretory protein 5	SVS5	−1.53876
Epsilon 2 globin	HBE2	−1.33216
Dehydrogenase/reductase 7	DHRS7	−1.28143
Keratin, type II cytoskeletal 5	KRT5	−1.20467
Bone marrow proteoglycan	PRG2	−1.05433
Uncharacterized protein	LOC100912707	−1.03433
Coronin 7	CORO7	−1.0073

## Data Availability

Data that support findings of this study are available upon request from the authors.

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
