# Peer review of "A Proteomic Screen to Unravel the Molecular Pathways Associated with Warfarin-Induced or TNAP-Inhibited Arterial Calcification in Rats"

_ijms, 2023, doi:10.3390/ijms24043657_

Round 1
Reviewer 1 Report
This concise report from Opdebeeck et al provides proteomic analysis of the effects of a TNAP inhibitor on a warfarin-treated rat model of medical arterial calcification. This is well written and is timely with high clinical impact.
Some clarifications:
Is the relationship to thermogenesis possibly related to the association of elite athleticism with coronary calcification?
Is the finding of the differential expression of the LXR/RXR pathway consistent with the finding of Hsu et al that LXR agonists promote SMC calcification?
The authors describe arterial calcification as life-threatening. But there is a small but growing trend in this field of research to suggest that arterial calcification is beneficial, in which case therapeutic agents to halt or reverse arterial calcification would not be necessary and possibly harmful. This notion seems to be based on the evidence that statin use (which is associated with CV benefit) is also associated with more rapid progression of coronary artery calcification; it is also based on the idea that hardening of arteries may “stabilize plaque.” Is there any evidence that stopping or reversing arterial calcification is protective? Or is there a suitable model to test this? It would be relevant to discuss these issues.
MINOR
Need to adjust commas to decimal points for this journal
Need to define abbreviation DEP (Differentially expressed proteins)
Reviewer 2 Report
This manuscript by Opdebeeck et al, described a proteomic screen using rats exposed to warfarin and treated with the TNAP-inhibitor SBI-425.
The story is interesting with potential and well-written but as presented seems only preliminary.
The proteomic screen has not received in vivo and molecular confirmation or validation. A single Western blot Figure (#1) is shown with non-significant data. This figure, which is clearly insufficient in itself, showed evocative but unsignificant data (stated in the text) but is later used in the discussed as validating data.
The M&M section indicated 10 rats were used but Fig 1 showed a n of 4. The large heterogeneity in the SBI-425 treatment data suggest either a poorly executed experiment or some other physiological effect that warrants investigation. This Fig 1 shows uncompleted work.
There is no other validation data with other western or protein-based or nucleic acid-based experiments.
The aorta is a large heterogeneous tissue, the arch is quite different more elastic, with high hemodynamic loads with qualitative and quantitative difference in cell population than the thoracic and abdominal aorta. There is no detail in the methods as to whether the entire aorta from the LV sinus to the iliac split was used or if a specific section was taken. Was the selection of the aorta chosen based on calcification? This should have been mentioned and taken into account as it could produce confounding data as calcification doesn’t occur uniformly along the aorta.
This whole manuscript seems to have been rushed and the experiments poorly considered.
A Figure 6 is mentioned in the discussion but there isn’t such figure or even a Fig 5. Does this reference a figure from another manuscript, the supplemental data ??
Reviewer 3 Report
Opdebeeck et al. investigated the factors related to the inhibition of arterial medial calcification through SBI-425 dosing. They showed that the actions of SBI-425 were associated with a significant downregulation of inflammatory signaling and LXR/RXR signaling, and an upregulation of mitochondrial metabolic pathways. The study is of interest, and the manuscript is well-written.
Author Response
I want to thank the reviewer for taking his/her time to read our manuscript. I am very pleased that he/she found the manuscript interesting and well-written.